# Light-Up Split Broccoli Aptamer as a Versatile Tool for RNA Assembly Monitoring in Cell-Free TX-TL Systems, Hybrid RNA/DNA Origami Tagging and DNA Biosensing

**DOI:** 10.3390/ijms24108483

**Published:** 2023-05-09

**Authors:** Emanuela Torelli, Ben Shirt-Ediss, Silvia A. Navarro, Marisa Manzano, Priya Vizzini, Natalio Krasnogor

**Affiliations:** 1Interdisciplinary Computing and Complex BioSystems (ICOS), Centre for Synthetic Biology and Bioeconomy (CSBB), Newcastle University, Newcastle upon Tyne NE1 7RU, UK; 2Dipartimento di Scienze AgroAlimentari, Ambientali e Animali, Università degli Studi di Udine, 33100 Udine, Italy

**Keywords:** light-up split Broccoli aptamer, RNA self-assembly monitoring, cell-free TX-TL system, hybrid RNA/DNA origami, target DNA detection, *Campylobacter* spp.

## Abstract

Binary light-up aptamers are intriguing and emerging tools with potential in different fields. Herein, we demonstrate the versatility of a split Broccoli aptamer system able to turn on the fluorescence signal only in the presence of a complementary sequence. First, an RNA three-way junction harbouring the split system is assembled in an *E. coli*-based cell-free TX-TL system where the folding of the functional aptamer is demonstrated. Then, the same strategy is introduced into a ‘bio-orthogonal’ hybrid RNA/DNA rectangle origami characterized by atomic force microscopy: the activation of the split system through the origami self-assembly is demonstrated. Finally, our system is successfully used to detect the femtomoles of a *Campylobacter* spp. DNA target sequence. Potential applications of our system include the real-time monitoring of the self-assembly of nucleic-acid-based devices *in vivo* and of the intracellular delivery of therapeutic nanostructures, as well as the *in vitro* and *in vivo* detection of different DNA/RNA targets.

## 1. Introduction

Light-up RNA aptamers offer an alternative to fluorescent probes and proteins (e.g., green fluorescent protein GFP), avoiding the covalent conjugation with fluorescent molecules and the alteration of natural proteins’ expression level and localization *in vivo* [1,2,3]. RNA aptamers able to bind a non-cytotoxic, cell-permeable and brighter dye (DFHBI and analogues) were successfully introduced and selected [4,5,6]. To further improve RNA imaging and aptamer stability in a cellular environment, Filonov et al. [7] obtained a new light-up aptamer; expressed in both prokaryotic and eukaryotic cells, Broccoli showed a high folding efficiency, a lower magnesium chloride dependence and an increased thermostability compared to Spinach and Spinach2 aptamers [8]. Because of this, Broccoli, either as full or split sequences, has received increased attention and found a variety of applications [1].

In this study, we investigate the versatility of our split system [9] in three different contexts. Specifically, we (i) monitor the three-way junction self-assembly in a cell-free system, (ii) develop a label for hybrid RNA/DNA origami and (iii) develop a detection system for a specific DNA sequence.

The split Broccoli concept was first introduced *in vivo* by dividing the full sequence into two autonomous RNA strands, called Top and Bottom; the two-strand system was genetically encoded and able to assemble in *Escherichia coli* [10]. However, the two aptameric fragments could not hybridize to a selected target complementary sequence, which can represent a limitation when self-assembly reaction monitoring is considered. Most recently, split versions of Broccoli were used *in vivo* as a reporter integrated into a catalytic hairpin assembly circuit [11], an aptamer-initiated fluorescence complementation assay [12] and an *in situ* amplification method [13].

In detail, the split system was integrated into a catalytic hairpin assembly circuit, in which the target triggered and catalysed the hybridization between two hairpins (H1 and H2) modified with split Broccoli, called Broc and Coli. However, once the aptamer was reconstituted, the target was disconnected [11], making the system unsuitable for stable nucleic acid self-assemblies (e.g., DNA origami [14] and RNA origami [15,16,17]). Following a different approach, the Broccoli reporter was combined with an RNA-based hybridization chain reaction for imaging RNA location: in the presence of an initiator RNA, H1 and H2 hairpins hybridized, resulting in a double-stranded concatemeric assembly able to activate a fluorescence signal [13], but potentially disturbing the cellular expression pattern. Considering a simpler design developed to image native mRNA in mammalian cells, Wang et al. [12] used two RNA split fragments extended with recognition sequences at the UUCG loop. Compared with the previous approach, this imaging strategy minimally disturbed the natural behaviour of mRNA, the cell morphology and the cell viability [12]. However, the system was not tested in prokaryotic cells, in which different transcribed terminators (e.g., a 47 nt T7 terminator instead of a 63 nt mini polyA terminator required by RNA pol II [12]) could result in different transcript composition and size, influencing a specific nucleic acid assembly.

Therefore, the described approaches [10,11,12,13] are unsuitable when the *in vivo* synthesis monitoring of stable DNA-encoded RNA origami [18] and memory structures [19] are considered as potential generic interfaces for bacterial cell processes (e.g., transcription, translation and transduction events). For this reason and as a further step during RNA origami synthesis [9,18] in bacteria, here, we first explore the use of our binary Broccoli [9,18] to monitor the RNA self-assembly in an *E. coli* transcription–translation (TX-TL) system, which represents an intermediate step before moving encoded RNA assemblies to synthetic cells or bacterial cells. The designed RNA three-way junction system including the split sequences was genetically encoded and expressed in an *E. coli* cell-free system at 37 °C: the self-assembly and the fluorescence emission signal were successfully simulated, visualized and detected.

Furthermore, we demonstrate the versatility and adaptability of the binary DFHBI-1T binding system combining the split functionality with a rectangle RNA/DNA hybrid origami nanostructure and suggesting its potential use as an alternative to organic dyes (e.g., Cy5 and Alexa488), which are covalently attached to single-stranded sequences and can lead to potential side effects on dynamic assemblies [20]. Furthermore, when *in vivo* self-assemblies are considered, the use of labelled sequences (with common Alexa and Cy fluorescent dyes) is unfeasible or requires invasive techniques (electroporation, micro-injection, transfection), while the cell-permeable and non-cytotoxic fluorophore DFHBI-1T allows the direct non-invasive fluorescence imaging of co-transcriptionally folded sequences. A ‘bio-orthogonal’ and uniquely addressable 982 nt RNA scaffold sequence is folded by short DNA staple strands into a rectangle origami characterized by atomic force microscopy (AFM). In this context, the use of a split system, instead of a full aptamer sequence, conjugated to two RNA staple strands opens up the possibility of monitoring folding and unfolding, where multiple orthogonal split systems are considered, and to introduce other functionalization triggered by the assembly, providing considerable potential for different applications.

We demonstrate the activation of the split system through the correct self-assembly of a well-defined rectangle shape (40 nm × 25 nm) hybrid RNA/DNA origami, able to fold isothermally at 53 °C within minutes.

Then, we demonstrate the application of the split Broccoli system for the sensitive *in vitro* detection of a specific DNA target sequence from the foodborne pathogen *Campylobacter* spp., mainly found in raw and undercooked meat. The two split sequences are conjugated into two distinct arms complementary to the *Campylobacter* spp. target sequence [21]. Upon the addition of the single-stranded DNA target, the two non-functional RNA sequences hybridize to form the functional binding site, thus switching on the fluorescence signal of the specific dye. We designed and investigated three different systems as shortened variants, where the linker between the arm and the aptamer fragment consists of different nucleotide lengths. Then, the selected system was successfully tested by two optimized, simple and rapid assays: in-gel imaging and *in vitro* fluorescence measurements at 37 °C.

Based on the above-mentioned results, for the first time, we successfully used a light-up split system to monitor the co-transcriptional RNA self-assembly in an *E. coli* PURExpress cell-free environment, to label RNA/DNA hybrid nanostructures and to detect the femtomoles of a specific DNA target at 37 °C in less than 20 min. Our findings expand the applications of split aptamers and show great potential in the tracking and biosensing of *in vitro*/*in vivo* assemblies.

Looking forward, our results pave the way to (i) the future development of sensitive aptameric optical biosensors as a potential alternative to RNA and DNA sensors [22,23,24,25] and (ii) protein-free tracking systems useful to monitor the stability, integrity and delivery of therapeutic hybrid RNA/DNA nanostructures used to deliver RNA scaffolds [26] (e.g., messenger RNAs) or siRNA in living cells. Moreover, DNA-encoded RNA assemblies, circuits and memory structures [19] carrying ‘light-up’ split aptamers can be designed, built and tested in cell-free systems as staging platforms before attempting engineering in living cells.

## 2. Results and Discussion

### 2.1. Expression and Self-Assembly in an E. Coli Cell-Free TX-TL System

Cell-free TX-TL systems and cell-free extracts are exciting tools that help characterize gene circuits and devices *in vitro* before moving *in vivo* for synthetic biology research. They are useful in gene network engineering, biomanufacturing and biosensing [27,28,29]. Furthermore, cell-free systems offer an intermediate step to prototype nucleic-acid-based materials [30] and RNA assemblies, before moving to the design of synthetic cells or more complex engineered cells.

The PURExpress system is one of the TX-TL platforms that combine the bacteriophage T7 RNA polymerase and T7 promoter with the translational machinery of *E. coli* [31]. In this paper, three components of a three-way junction carrying the split system were encoded in the corresponding dsDNA gBlocks containing the T7 promoter and T7 terminator (Appendix A). After amplification and purification, all the purified templates were transcribed, purified and checked by denaturing PAGE. As each transcript showed the expected size (Appendix A), the self-assembly at 37 °C of purified RNA sequences (Figure 1) was at first demonstrated by in-gel imaging (Figure 2).

The gel image (Figure 2) shows the selective DFHBI-1T staining of the three-way junction (lane 7) and the full Broccoli aptamer (positive control, lane 2), due to the aptameric G-quadruplex.

Computational prediction of the Broccoli three-way junction showed that the assembly was predicted even when T7 terminator sequences were included at the 3′ end of all RNA strands: the complex is formed when the concentrations of Split1d and Split2d are equal to or greater than the concentration of the complementary strand (Appendix A). Furthermore, heatmaps underlined that Split1d and Split2d are not able to hybridize without the complementary sequence (Appendix A). Split1d/complementary and Split2d/complementary complexes exist as reported in Appendix A. The minimum free energy (MFE) structure of the full Broccoli complex is shown in Appendix A. The MFE state is predicted to contain the Broccoli aptamer region, even when T7 terminator sequences are included at the 3′ end. When we considered the split system previously reported [12] as a component of the three-way junction with T7 terminators, the full aptamer reconstitution was not fully predicted (Appendix A).

Finally, the three-way junction was expressed from double-stranded templates and folded in the PURExpress system, enabling the fluorescence activation derived from the functional RNA Broccoli split system. As reported in Figure 3, after about 25 min at 37 °C the full assembly showed a fluorescence intensity of 1.84 (±0.39, 3 independent measurements), while all the other samples (Split1d, Split2d, complementary sequence, Split1d or Split2d and complementary sequence, Split1d and Split2d) showed a fluorescence signal comparable to the basal fluorescence level of the cell-free system, underlining that the aptamer was not assembled. To further confirm the self-assembly of the three strands in the cell-free TX-TL system, aliquots were run on 10% TBE gel. As shown in Appendix A, the three-way junction assembled in the cell-free system (lanes 6, 7, 8 and 9, black arrow) is characterized by the same migration distance of the three-way junction assembled from transcribed and purified RNA (lane 3).

### 2.2. Activation of the Split Broccoli Aptamer through Hybrid RNA/DNA Origami

Scaffolded DNA origami technology enables the bottom-up synthesis of nanostructures of defined shape and dimension, characterized by different functionalities and applications [32]. In detail, a single-stranded DNA (ssDNA) scaffold sequence is folded into a precise geometry by several short ssDNA sequences, called staple strands [33].

While many 2D and 3D DNA origami have been designed and visualized, hybrid RNA/DNA origami remains less explored despite the great potential to introduce functional RNA (e.g., RNA aptamers or siRNA) or mRNA as scaffolds [34,35,36,37,38].

Recently, we proposed a custom-made software to select de Bruijn sequences (DBS of order 6) to be used as scaffolds in origami synthesis [36]. Specifically, DBS of order *k* have no duplicate subsequences of size *k* or larger and thus are in theory uniquely addressable. The obtained scaffolds are uniquely addressable and ‘bio-orthogonal’ by design and can be successfully folded into DNA or RNA origami [9,18] and RNA/DNA hybrid origami [36].

Here, we designed a hybrid ‘bio-orthogonal’ RNA/DNA origami rectangle (Figure 4) with dimensions of 40 nm × 25 nm using caDNAno [39]. More specifically, the DBS scaffold had no repeats longer than 6 nt and excluded undesirable biological sequences, including the start codon, Shine–Dalgarno sequence and restriction enzyme sites. A de Bruijn sequence of order *k* = 6 has a sliding window of length 6 nt that is never repeated as it is moved along the sequence [36].

The RNA scaffold (DBS 982) was synthesized by T7 transcription from a purified amplification product from plasmid pUCIDT-AMP-T7p-DBS982 (Appendix A): after denaturing PAGE, the purified transcript showed the expected size (Appendix A). The hybrid origami was then folded into a rectangle using 31 synthetic DNA staple strands (Appendix A) and following a rapid isothermal protocol (53 °C for 15 min). To preliminarily check the folding quality, the unpurified and purified RNA/DNA origami were run on 1.5% agarose gel. After electrophoresis, the gel images showed two distinct main bands for both unpurified (Appendix A) and purified (Appendix A) samples, suggesting proper self-assembly. In detail, the faster band corresponded to the RNA/DNA origami and showed a slightly different migration distance compared to the scaffold as underlined by the black dashed line (Appendix A). We hypothesized that the slower band corresponded to nonspecific dimers, which were frequently observed by gel electrophoresis [40] and AFM [33], and appeared to be the result of stacking interactions as reported [33,40]. The purified sample was visualized by AFM: the AFM images confirmed the correct formation of the origami with geometry and lengths consistent with the expected design (Figure 4, Appendix A). Nonspecific monomer interactions were also imaged by AFM (Appendix A).

Finally, the split system was integrated into the hybrid origami characterized by AFM (Appendix A). To demonstrate the functional activation of the split system through origami self-assembly, samples were run in an agarose gel, stained with DFHBI, washed and stained with SYBR^®^ Gold. In previous work [41], to verify the split malachite green aptamer activation through the self-assembly of an RNA cube characterized by cryo-EM, PAGE and fluorescence measurements were considered instead of direct imaging. It should be noted that one dsDNA target [42] or an individual aptamer [43] protruding from 2D nanostructures were not clearly visible by AFM [42,43], probably due to the orientation of the extended individual split aptamer staple strands with respect to the scanning direction during liquid-mode imaging (under buffer) [43]. Instead of considering an individual protruding double-stranded sequence, several staple strands with dumbbell hairpin labels were used as markers for AFM contrast to create patterns on 2D DNA origami: the imaged patterns were made by adjacent packed hairpins [33]. During high-resolution AFM imaging, poorly patterned images were also obtained due to tip-induced damage or flipped structures with dumbbell hairpins down. The author also noted that during an AFM session, only 1 in 10 AFM tips could reveal fine structure and distinguish individual hairpin labels. Additionally, a second type of explored duplex marker was even more difficult to image due to its flexibility [33].

In our design, two DNA staples were converted to RNA sequences and elongated in 5′ and 3′ with Split1 and Split2. First, to demonstrate the fluorescence emission due to the hybridization between elongated staples S1 and S2 and the complementary 36 nt short RNA derived from the DBS scaffold, we used specific DFHBI-1T in-gel imaging, a simple, selective and sensitive assay introduced by Filonov et al. [8] to monitor different expressed RNAs tagged with Broccoli. As shown in Appendix A, after the hybridization reaction and in-gel staining with DFHBI-1T, the assembly between modified staples S1 and S2 and the complementary DBS showed a clear fluorescent band (lane 7), thus demonstrating the ability of the system to turn on the fluorescence of the specific dye. Then, the split system was included in the hybrid origami: the purified origami sample was run on 1.5% agarose gel, with scaffold and staple strands as negative controls. After electrophoresis, the agarose gel was stained with DFHBI-1T and a fluorescent band related to the formation of the aptamer was imaged (Figure 5 left, lane 4, green arrow) at the migration distance corresponding to the purified hybrid origami (Figure 5 right, lane 4), while the scaffold sequence and staple strands were unable to emit a fluorescence signal (Appendix A). It should be noted that compared to the previously described in-gel imaging using polyacrylamide gels, agarose gel staining with DFHBI-1T requires a longer incubation time and higher PMT voltage (800 V) during laser scanning.

These results suggest that the split system can find potential application in studies related to the *in vivo* delivery and tracking of hybrid origami using a non-cytotoxic dye. Furthermore, multiple split systems [41] can be introduced in the same design with nanometre precision, to increase the DFHBI-1T emission signal and to monitor the retention of folding, thus reporting on the stability and integrity of assemblies during internalization.

### 2.3. Sequence Design, Dot Blot Assay on Target CP3 and Denaturing PAGE Analysis of DNA/RNA Oligonucleotides

To further confirm the versatility of our split system and investigate its ability to perform DNA detection, the split binding arms were changed to be specific to a DNA target from *Campylobacter* spp. [21]. We considered our split system and two analogues with a shorter spacer arm of four base pairs or without a spacer. Indeed, as previously described, the choice of the split site and the linkers between the affinity arms and the aptameric fragment are key factors to be considered when designing a binary aptamer for applications in diagnostics [1,44]. Chandler et al. (2018) [45] designed and experimentally tested different sets of the F30-Broccoli split considering the two halves’ sequences and moving the location of the split. The results reveal the importance of the split site, which defines the fluorescence activation (or lack thereof) due to the assembly reaction. Following a different strategy, in our design, the split sequences should emit a fluorescent signal only in the presence of a DNA target sequence (Figure 6).

In detail, the split sequences Split1 and Split2 were conjugated in the 5′ or 3′ end with RNA sequences complementary to a selected DNA target (CP3, 36 nt) from *Campylobacter* spp. (Patent Number IT 102020000012496 and Patent Number FR 2005578) [21,46,47] (Appendix A), the most common bacterial cause of human gastroenteritis in the world [48]. The 36 nt (18 nt on each split sequence) oligonucleotide extension was designed on the 16S rRNA gene encoding for the ribosomal *Campylobacter* RNA of *C. jejuni*, *C. coli*, *C. lari* and *C. upsaliensis*. The negative controls PE and PR corresponded to a 36 nt *E. coli* sequence characterized by similarities to *Campylobacter*, and a 36 nt sequence designed by mismatching positions of the target sequence [21], respectively, resulting in a partial complementarity (12 bp) with the Split1 sequence. The complementary biotinylated DNA detection probe (CampyP3) was used in a dot blot assay, as previously described [21]: the obtained results showed a probe sensitivity of 1 ng/µL (Appendix A). The probe (Patent Number IT 102020000012496 [46] and Patent Number FR 2005578 [47]) specificity was underlined in Vizzini et al. (2021) [21].

Since RNA and DNA sequences were used at specific molar concentrations and ratios, the purity of the designed split sequences, target CP3 and negative controls PE and PR [21] (Appendix A) was checked by denaturing PAGE (Appendix A), loading ~25 ng for each sample. The results show that RNA oligos, especially longer Split2 sequences, are characterized by a lower purity compared with DNA sequences. As previously reported, the synthesis and purification of synthetic RNA oligonucleotides are less efficient compared with deoxynucleotides due to the 2′-OH protection, and the formation of several conformations (e.g., related to short intra-molecular double-stranded regions or stem–loop folding), especially for RNA longer than 50 nt [49]. Nonetheless, considering the low micromolar concentrations (120 nM corresponding to ~8 ng) to be used in the following in-gel imaging experiments, the RP-HPLC purified synthetic RNA oligos were not further purified.

Finally, the positive control Broccoli aptamer was transcribed from an amplified and purified dsDNA with a T7 promoter: the transcript was analysed by denaturing PAGE and the expected size was confirmed (66 nt, Appendix A).

### 2.4. Split Broccoli Aptamer System: PAGE Analysis and in-Gel Imaging

To investigate the different split aptamer systems, Split1 and Split2 were mixed with CP3 (positive control), PE and PR (negative controls). After incubation at 37 °C for 25 min in the hybridization buffer supplemented with 10 mM MgCl_2_, samples were analysed by 10% PAGE. Split1/Split2-4nt, Split1/Split2-8nt and Split1/Split2-0nt showed an upper clear band, smeared and multiple bands or a faint band, respectively (Appendix A). For these reasons, Split1/Split2-4nt was selected to run the in-gel imaging experiment to confirm that the above-mentioned upper band was the reconstituted aptamer upon CP3 hybridization (Figure 6b). After electrophoresis, the 6% polyacrylamide gel was stained with 0.5 µM DFHBI-1T solution for 4 min. The gel imaging resulted in the selective staining of the hybrid assembly Split1/Split2-4nt/CP3 (Figure 7a, lane 5, green arrow), while all the other loaded samples (Split1/Split2-4nt, CP3, PE, PR, Split1/Split2-4nt in the presence of PE or PR) were negative (Figure 7a, lanes 3, 4, 6, 7, 8 and 9). It should be noted that a higher DFHBI-1T concentration and longer incubation time may result in undesirable non-specific staining. After fluorophore dye staining, the gel was washed three times and stained with SYBR^®^ Gold to visualize RNA and DNA bands (Figure 7b).

Finally, the double staining technique was applied to analyse samples with the same equimolar concentration of Split1/Split2-4 nt (0.12 µM) and reduced CP3 target concentration (0.10 µM, 0.08 µM and 0.05 µM). The lower detected CP3 concentration using DFHBI-1T corresponded to 0.6 ng/µL (Appendix A). This result underlines that the simple in-gel imaging technique can be used as a rapid investigation tool when assemblies are considered even at very low target concentrations, thus demonstrating its sensitivity.

In previous studies [45,50], in-gel imaging or fluorescence measurements were successfully considered as experimental procedures to test different Broccoli split florets. Two Broccoli halves (called Broc and Coli) were rationally designed and experimentally evaluated. Selected split sequences were then tested under different stimuli. In detail, native-PAGE gels stained with DFHBI-1T were used to evaluate different Broc and Coli sequences and their fluorescent behaviour in the presence of enzymes or of a fully complementary DNA oligonucleotide.

### 2.5. Split Broccoli Aptamer System: Fluorescence Measurements In Vitro

The crystal structure of the Spinach-DFHBI complex revealed that potassium ions are part of the folded structure and stabilize the G-quadruplex, activating the fluorescence [51]. As the Broccoli aptamer retains the G-quadruplex of the DFHBI-binding site of the Spinach aptamer, we postulated that the Broccoli-DFHBI complex is characterized by a similar structure. Recently, we compared the fluorescence values of the Broccoli aptamer in 20 mM Tris HCl pH 7.6, 1 mM EDTA, 10 mM MgCl_2_, and in 40 mM HEPES, 1 mM MgCl_2_, decreasing the potassium ion concentrations (from 100 mM to 0 mM) [9]. Reducing the K^+^ concentration in both buffers resulted in a gradually lower fluorescence emission intensity with a higher emission at 100 mM and 50 mM KCl [9].

The above-described in-gel imaging protocol was performed in DFHBI-1T aptamer buffer (40 mM HEPES pH 7.4, 100 mM KCl, 1 mM MgCl_2_ and 0.5 μM DFHBI-1T), while the hybridization between the split system and target CP3 was conducted in the hybridization buffer containing 10 mM MgCl_2_, a typical magnesium ion concentration for the stability and folding of RNA in molecular biology experiments [52].

Before moving from in-gel imaging to *in vitro* experiments, we first analysed by PAGE the hybridization products derived from reactions run in two different buffers: (a) DFHBI-1T aptamer buffer without MgCl_2_ or containing 5 mM or 10 mM MgCl_2_; and (b) hybridization buffer without KCl or supplemented with 50 mM or 100 mM KCl. Indeed, the split system performance should present a high emission fluorescence intensity signal and a low background, correlated with DFHBI complex stability (K^+^ dependent) and low Split1/Split2-4nt partial folding in the absence of the target CP3 (Mg^2+^ dependent), respectively. It should be noted that NUPACK analysis [53] showed stable Split1-4nt and Split2-4nt secondary structures and a very low equilibrium concentration of Split1/Split2-4nt (0.00024 µM), considering the default RNA setting for salts (1 M Na^+^ and 0 M Mg^2+^), an incubation temperature of 37 °C and a concentration of 50 nM for each sequence (Appendix A).

In the DFHBI aptamer buffer, the Split1/Split2-4 nt/CP3 hybrid band intensity increased considering the higher concentration of MgCl_2_, as expected (Appendix A). Overall, the band intensities were lower compared with the band intensities related to the same reaction conducted in the hybridization buffer (Appendix A). In buffers containing MgCl_2_ (5 mM or 10 mM), a faint band with the same migration distance appeared in samples Split1/Split2-4nt, Split1/Split2-4nt/PE and Split1/Split2-4nt/PR, presumably due to a low partial hybridization between Split1-4nt and Split2-4nt (Appendix A). Considering the results from PAGE analysis and to ensure a single predominant assembly in solution (Split1/Split2-4nt/CP3), the folding reactions and *in vitro* fluorescence measurements were initially performed in 20 mM Tris HCl pH 7.6, 1 mM EDTA, 100 mM KCl and 10 mM MgCl_2_ supplemented with DFHBI-1T.

After incubation at 37 °C for 15 cycles, the raw fluorescence values were measured by qPCR equipment. The Mg^2+^ concentration was reduced from 10 mM to 2.5 mM to further guarantee a weak hybridization between Split1 and Split2. In the optimized buffer, Split1/Split2-4nt/PE and Split1/Split2-4nt/PR showed values close to Split1/Split2-4nt, while Broccoli (17 ng) and Split1/Split2-4nt/CP3 (0.05 µM equimolar concentration) showed high fluorescence signals (Figure 8). The split system including the CP3 probe (5′-TAGTGGCGCACGGGTGAGTAAGGTATAGTTAATCTG-3′) was able to discern among complementary sequences (CP3), non-complementary sequences (PE), and partially complementary sequences (PR) designed by mismatching the positions of the target sequence; when PE or PR was used as a negative control, the fluorescence was not detected.

The *in vitro* photostability of the split system was assessed: after incubation at 37 °C for 15, 30 and 45 cycles, the fluorescence intensity values were 0.94 ± 0.1, 0.85 ± 0.07 and 0.93 ± 0.01, respectively (raw data blank subtracted, mean ± standard deviation, *n* = 3). The results underline that the Split1/Split2-4nt/CP3 complex was photostable.

The sensitivity curve was obtained by subtracting the background fluorescence (Split1/Split2-4 nt) from the raw fluorescence data for different CP3 concentrations from 0.12 ng/µL to 0.57 ng/µL (three independent measurements each, Figure 9). The correlation coefficient R^2^ was 0.995 and the limit of detection (LOD) was 0.074 ng/µL. The linear curve function was f(x) = 1.192x + 0.003 (Figure 9) and its slope represented the sensitivity of the split system.

## 3. Materials and Methods

### 3.1. Reagents and Materials

All RNA oligonucleotides were purchased from Eurogentec and resuspended in Ultra Pure^TM^ DNase/RNase-free distilled water to give stock solutions of 100 μM and stored at −80 °C. DNA oligonucleotides were purchased from Eurofins Genomics, Eurogentec or IDT. The pUCIDT-AMP-T7p_DB982 plasmid was purchased from IDT. The DNA oligonucleotides were resuspended in Ultra Pure^TM^ DNase/RNase-free distilled water to give stock solutions of 100 μM and stored at −20 °C. The pUCIDT-AMP-T7p_DB982 plasmid was resuspended in 40 µL of IDTE buffer (10 mM Tris, 0.1 mM EDTA) and stored at −20 °C. Ultra Pure^TM^ 10× Tris Borate EDTA (TBE) buffer, 0.5 M EDTA pH 8.0, Ultra Pure^TM^ 1 M Tris-HCl pH 7.5, Ultra Pure^TM^ DNase/RNase-free distilled water, 5 M NaCl (0.2 μm filtered), SYBR^®^ Gold, 10× Tris Acetate EDTA (TAE) buffer molecular biology grade RNase-free, 6% Novex^TM^ TBE gel, 10% Novex^TM^ TBE gel and 10% Novex^TM^ TBE-Urea gel were purchased from Thermo Fischer Scientific. Nickel chloride, 1 M HEPES pH 7.0 Bioreagent, 1 M KCl BioUltra, 1 M MgCl_2_ BioUltra, agarose and Nancy-520 were purchased from Sigma-Aldrich (St. Louis, MO, USA).

(5*Z*)-5[(3,5-Difluoro-4-hydroxyphenyl)methylene]-3,5-dihydro-2-methyl-3-(2,2,2-trifluoroethyl)-4*H*-imidazol-4-one (DFHBI-1T) was purchased from Tocris Bio-techne. A DFHBI-1T stock solution (20 mM) was prepared in DMSO (Sigma-Aldrich), stored in the dark at −20 °C and used within 4 weeks.

### 3.2. Sequence Design and Prediction of Complex Folding

Three RNA sequences [9] were considered as components of a three-way junction and were cloned into dsDNA gBlocks containing T7 promoter and T7 terminator (Split1d, Split2d and complementary sequence): all the RNA sequences were analysed using NUPACK software v4.0.0.27 [53] and used in cell-free TX-TL experiments.

The uniquely addressable and bio-orthogonal synthetic RNA scaffold was generated with the computer code presented by Kozyra et al. [36]. The workflow for bio-orthogonal DBS generation included the construction of the de Bruijn graph, filtering of biological sequences and further optimization to a particular specification (e.g., weaker secondary structure) [36]. The rectangle RNA-DNA hybrid origami (40 nm × 25 nm) was designed using caDNAno v2 and scadnano software as described [33,39,54,55].

Split sequences (Split1/2-8nt) ending with 8 nt of a stabilizing bio-orthogonal scaffold [8] were elongated in 5′ and 3′ with two sequences complementary to the CP3 target, while the 4 nt terminal stem–loop UUCG was removed as previously described [9]. Split1/2-4nt and Split1/2-0nt showed a shorter or absent F30 arm, respectively. Split1 and Split2 are the two split Broccoli fragments.

Our NUPACK predictions for the minimum free energy structure were run with RNA material for all strands, even when the target strand was DNA. Considering that NUPACK does not have an energy model for DNA–RNA hybrid complexes, we opted to assign all strands as RNA since the region with the most complex secondary structure (the Broccoli region) is RNA. Additionally, our NUPACK predictions assume 1M Na+, 0M Mg2+ salt conditions because salt correction terms can only be specified in NUPACK for pure DNA complexes. As such, our simulation results should be viewed as providing just an estimation of binding effects which should be interpreted in conjunction with the experimental results we present.

The RNA split sequences were analysed using NUPACK software [53]: the analysis was set at 37 °C, at a concentration of 50 mM for each RNA sequence and considering the salt concentration as 1 M Na^+^ and 0 M Mg^2+^ (default setting for RNA).

The target sequence CP3 (36 nt) was designed considering the 16S gene encoding ribosomal *Campylobacter* RNA of *C. jejuni*, *C. coli*, *C. lari*, and *C. upsaliensis* (base location: 72–108), as previously reported [21,46,47]. In detail, the probe sequence was designed considering sequences downloaded from GeneBank: the alignment was conducted using the online tool multiple sequence alignment with hierarchical clustering [56]. The selected sequence features were analysed by Amplifix 1.7.0 [57] and then by IDT OligoAnalyzer3.1 (https://eu.idtdna.com/calc/analyzer, accessed on 1 March 2019) [58]. The probe must fall within the following optimal parameters: melting temperature 60–80 °C and GC% content 40–60%. Moreover, the setting up of no GC clamp, no secondary structures (hairpins), no self-dimers or homodimers, no repeats and 3′ end stability was considered. The sequence of the selected probe with the optimal features was tested by in silico analysis to verify its specificity using FastPCR6.1 [59] and Amplifix 1.7.0 [57] software. A panel of different species of *Campylobacter*, together with several microorganisms commonly found in food samples and bacteria with high genomic similarity, were tested (Appendix A).

Finally, two ssDNA sequences, called PR and PE, of the same length as CP3 were used as negative controls. PR was designed by mismatching the positions of nucleic acids of the CP3 sequence, while PE corresponded to the sequence of *E. coli* (accession number 527445.1, base location: 338–376 for *E. coli*), which showed some similarities with *Campylobacter* [21].

### 3.3. Expression, Assembly and Fluorescence Measurements in PURExpress Cell-Free TX-TL System

The dsDNA gBlock templates (Broccoli, Split1d, Split2d and complementary sequence) containing the T7 promoter and terminator were amplified using the specific primers (0.25 µM or 0.5 µM) and Phusion^®^ DNA polymerase (NEB). The amplification program was as follows: 98 °C for 30 sec followed by denaturation at 98 °C for 10 sec, annealing for 15 sec and extension at 72 °C for 15 sec (15 cycles), additional extension for 5 min at 72 °C. The annealing temperatures were 68 °C (Broccoli), 67 °C (Split1) or 69 °C (Split2 and complementary sequence). After purification using Monarch^®^ PCR & DNA Cleanup kit (NEB), the templates were transcribed, and the purified transcripts were quantified and checked using the NanoDrop One/OneC spectrophotometer and 10% TBE-Urea gel, respectively.

In order to verify the assembly products, RNA Split1d, Split2d and complementary (0.12 µM) were incubated at 37 °C for 25 min in 20 mM Tris, 1 mM EDTA, 100 mM KCl and 2.5 mM MgCl_2_, and then analysed by PAGE and in-gel imaging. All samples were run on Novex^TM^ TBE gel in 1× TBE buffer. In-gel imaging was performed as previously described [8] with some modifications. Briefly, the gels were washed three times for 5 min in RNase-free water and then stained for 2–3 min in DFHBI aptamer buffer containing 40 mM HEPES pH 7.4, 100 mM KCl, 1 mM MgCl_2_ and 0.5 μM DFHBI-1T (50 mL of buffer in a 12 cm square Petri dish, Greiner Bio-One Ltd.). The gels were imaged using a Typhoon laser scanner (excitation 488 nm, emission 532 nm, normal sensitivity and PMT 380 V). Then, the gels were washed three times with RNase-free water, stained with SYBR^®^ Gold in 1× TBE for 5 min, and visualized using a Typhoon laser scanner. The low-range ssRNA ladder (NEB) was used as a molecular weight marker.

Cell-free experiments were performed using the PURExpress^®^ system (NEB), following the manufacturer’s protocol. Briefly, 8 µL of solution A and 6 µL of solution B were mixed with 16 U RNase Inhibitor (NEB), gBlock templates (2 nM each) and 20 µM DFHBI-1T. The final reaction volume was adjusted to 20 µL with nuclease-free water. All reactions (3 replicates for each combination) were set up on ice, avoiding bubbles, and incubated at 37 °C for about 25 min in a PCR cycler Rotor-Gene Q (Qiagen, excitation at 470 nm, emission at 515 nm, gain 10). It should be noted that special care is fundamental to avoid air bubbles, which are a source of variability and may interfere with fluorescence measurements, increasing the possibility of outliers, a potential problem in cell-free experiments [29].

Finally, to further confirm the full assembly of the 3-way junction in the PURExpress system, after incubation at 37 °C and fluorescence measurements using the qPCR thermal cycler, aliquots (2.5 or 3.5 µL) from the 20 µL cell-free reaction mix (positive samples) were run on 10% TBE gel at 200 V for 1 h and 10 min and compared with the assembled purified transcript described above. All samples were stained with SYBR^®^ Gold in 1× TBE for 25 min.

### 3.4. Hybrid RNA/DNA Origami Synthesis and Purification

A double-stranded DNA scaffold template was obtained from pUCIDT-AMP-T7p_DB982 plasmid (10 ng in 50 µL reaction mixture) via PCR amplification using Phusion^®^ DNA polymerase (NEB), T7-DB982 forward and T7-DB982 reverse primers (0.5 µM each). An initial denaturation at 98 °C for 30 s was followed by denaturation at 98 °C for 10 s, annealing at 68 °C for 15 s and extension at 72 °C for 15 s (30 cycles). Finally, an additional extension was achieved for 10 min at 72 °C. The PCR product was purified using a Monarch^®^ PCR & DNA Cleanup kit (NEB) and eluted in 10 µL of Ultra Pure^TM^ DNase/RNase-free distilled water. The purified amplicon and the 1kb DNA ladder (NEB) were run on 1% agarose gel in 1× TBE for 1 h 30 min at 110 V. The gel was pre-stained with Nancy-520 and visualized under UV illumination (UVP GelStudio, AnalitikJena). The DNA concentration was measured on a NanoDrop One/OneC spectrophotometer (Thermo Scientific). The purified template was transcribed *in vitro* at 37 °C for 1 h using an Ampliscribe^TM^ T7-Flash^TM^ Transcription kit (Lucigen, Epicentre) following the manufacturer’s instructions. After DNase treatment at 37 °C for 15 min, the RNA transcript was purified using RNA Clean & Concentrator^TM^ (Zymo Research), eluted in 10 µL of Ultra Pure^TM^ DNase/RNase-free distilled water and quantified using a NanoDrop spectrophotometer. After the addition of the RNA loading dye 2× (NEB) and the heat denaturation at 65 °C for 5 min, the RNA scaffold transcribed *in vitro* was loaded and run on 10% TBE-Urea gel in TBE buffer at 200 V for 45 min. After staining with SYBR^®^ Gold in 1× TBE for 5 min, the gel was visualized using a Typhoon laser scanner. The low-range ssRNA ladder (NEB) was used as a marker of molecular weight.

Single-stranded DNA staple strands (Appendix A; final concentration 500 nM each) were mixed in a 50-fold excess with an RNA scaffold in 50 μL of folding buffer (1× TAE molecular biology grade, 40 mM Tris-acetate and 1 mM EDTA, supplemented with 12.5 mM MgCl_2_). The folding solution was incubated at 53 °C for 15 min and then held at a constant temperature of 4 °C to stop the reaction. The folded constructs were purified from an excess of staple strands using the Amicon Ultra 0.5 mL 100 kDa centrifugal filters (Millipore). Capped Amicon Ultra was rinsed with 500 µL of RNase-free folding buffer and centrifuged at 14,000× *g* for 1 min at 11 °C. After this preliminary washing step, the RNA/DNA origami samples were added to the filter device. Between every centrifugation step (14,000× *g* for 1 min at 11 °C repeated 5 times), the flowthrough was removed, and the filter was refilled with 450 μL of folding buffer. To recover the purified and concentrated origami sample, the filter was turned upside down and centrifuged at 1000× *g* for 5 min at 11 °C. The scaffold, staple strands and unpurified and purified RNA/DNA origamis were run on 1.5% agarose gel in 1× TAE buffer at 90 V for 1 h at low temperature (below 10 °C). After staining with SYBR^®^ Gold in 1× TAE for 8 min, the gels were visualized using a Typhoon laser scanner.

### 3.5. Hybrid DNA/RNA Origami Characterization: Atomic Force Microscope (AFM) Imaging

Freshly cleaved mica (Mica Grade V-4 12 mm Discs x 0.15 mm, Azpack Ltd., Loughborough, UK) was passivated for 20 sec with 20 μL of 10 mM NiCl_2_ in Ultra Pure^TM^ DNase/RNase-free distilled water to ensure the adhesion of the negatively charged origami structures on the negative mica surface. After three washing steps with folding buffer (60 μL each), the purified origami solution was diluted 1:2 in the same buffer, added (10 μL) to the passivated mica surface, allowed to absorb for 1 min and imaged immediately (~100 μL of the folding buffer was added after the origami absorption).

The purified hybrid RNA/DNA origami was characterized in tapping mode in liquid using VRS AFM Cypher ES (Asylum Research, Oxford Instruments, Santa Barbara, CA, USA). The vertical oscillation of the BioLever Mini tip (spring k of 0.09 N/m, Asylum Research, Oxford Instruments, Santa Barbara, CA) was controlled by photothermal excitation (Blue Drive). All the images were lightly corrected using Gwyddion software v2.56 [60].

### 3.6. Hybrid RNA/DNA Origami and In-Gel Imaging

Two DNA staple strands were replaced with two protruding RNA staple strands including Broccoli split aptamer sequences. The hybridization reaction was run on 10% Novex^TM^ TBE gel and analysed by in-gel imaging as previously described. Then, to confirm the incorporation of the split Broccoli aptamer system into the purified rectangle nanostructure, the hybrid RNA/DNA origami was run on 1.5% agarose gel in TAE buffer (75 mL was poured into a 15 cm × 10 cm gel tray, Biorad) and analysed by in-gel imaging using the specific dye DFHBI-1T as described above, with a few modifications due to the different gel composition (agarose instead of polyacrylamide) and thickness. Briefly, the gel was washed three times for 5 min in RNase-free water and then stained in a 12 cm square Petri dish for 1 h and 40 min with 50–60 mL of aptamer buffer containing 0.5 μM DFHBI-1T. The gel was imaged using a Typhoon laser scanner (excitation 488 nm, emission 532 nm, normal sensitivity and PMT 800 V). Then, the gels were washed three times with RNase-free water, stained with SYBR^®^ Gold in 1× TAE for 10 min, and visualized using a Typhoon laser scanner.

### 3.7. Dot Blot Assay

The following samples were spotted onto the positively charged nylon membrane (Hybond^TM^ XL, GE Healthcare, Chicago, IL, USA) target CP3 (1 μL) at 100 ng/μL, 50 ng/μL, 10 ng/μL, 1 ng/μL, 0.1 ng/μL, 0.01 ng/μL, 1 pg/μL and 0.1 pg/μL, to check the sensitivity as previously described [21]. Briefly, before deposition on membranes, DNA samples were denatured at 95 °C for 10 min and chilled immediately on ice. All samples (1 μL) were spotted on the nylon membranes and cross-linked to the air-dried membrane by exposure to UV light for 10 min. The membranes were soaked in a pre-warmed hybridization buffer at 65 °C for 30 min under gentle shaking. Hybridization was carried out at 65 °C overnight in the same buffer supplemented with 4 ng/μL of denatured biotinylated CampyP3 probe (100 ng/μL). After incubation, the membranes were washed twice with 300 mM saline sodium citrate (SSC), 0.1% SDS for 5 min at room temperature on a shaker and then with 75 mM SSC for 15 min. The membranes were incubated in a blocking solution at room temperature for 15 min and finally incubated with the same blocking solution supplemented with 0.7 μM streptavidin-HRP (30 min at room temperature). The signal was revealed using the enhanced chemiluminescent substrate for the detection of HRP (Thermo Fisher Scientific, Waltham, MA, USA) under a ChemiDoc MP imaging system [21].

### 3.8. Synthesis of Broccoli Aptamer and Purity Control of the Split Aptamer Sequences (Split1,2-8nt, Split1,2-4nt and Split1,2-0nt) by Denaturing PAGE

The double-stranded DNA template to be transcribed containing the T7 promoter sequence was prepared by polymerase chain reaction (PCR) from a Broccoli single-stranded DNA sequence (100 ng in 20 µL reaction mixture) amplified using Broccoli forward and reverse primers (0.5 µM) and Phusion^®^ DNA polymerase (NEB). An initial denaturation at 98 °C for 30 s was followed by denaturation at 98 °C for 10 s, annealing at 64 °C for 20 s and extension at 72 °C for 15 s (20 cycles). Finally, an additional extension was achieved for 5 min at 72 °C. The PCR product was purified using Monarch^®^ PCR & DNA Cleanup kit (NEB) and eluted in 10 µL of Ultra Pure^TM^ DNase/RNase-free distilled water. The purified amplicon and the low-molecular-weight DNA ladder (NEB) were run on 2% agarose gel in 1× TBE for 1 h 40 min at 100 V. The gel was pre-stained with Nancy-520 and visualized under UV illumination (UVP GelStudio, AnalitikJena). The DNA concentration was measured on a NanoDrop One/OneC spectrophotometer (Thermo Fisher Scientific, Labtech International Ltd., Heathfield, UK).

The purified template was transcribed *in vitro* at 37 °C for 1 h and 30 min using Ampliscribe^TM^ T7-Flash^TM^ Transcription kit (Lucigen, Epicentre). After DNase treatment at 37 °C for 15 min, the RNA transcript was purified using RNA Clean & Concentrator^TM^ (Zymo Research) and eluted in 10 µL of Ultra Pure^TM^ DNase/RNase-free distilled water and quantified using a NanoDrop spectrophotometer.

After the addition of the RNA loading dye 2× (NEB) and the heat denaturation at 65 °C for 5 min, the Broccoli aptamer transcribed *in vitro* was loaded and run on 10% TBE-urea gel in TBE buffer at 200 V for 45 min. After staining with SYBR^®^ Gold in 1× TBE for 5 min, the gel was visualized using a Typhoon laser scanner and Image Quant TL software v10.2 (normal sensitivity and PMT 380 V; GE Healthcare Life Sciences). The low-range ssRNA ladder (NEB) was used as a marker of molecular weight.

To check the synthesis purity, the chemically synthesized RNA and DNA sequences (split aptamer sequences, target CP3 and the non-complementary sequences PE and PR) were analysed by 10% TBE-Urea gel electrophoresis following the protocol described above.

### 3.9. Assembly of RNA Split Sequences with Complementary CP3 and Non-Complementary Sequences PE and PR: PAGE and In-Gel Imaging

All the RNA and DNA sequences were diluted 1:10 from 100 μM stock solution. Each solution was quantified (ng/μL, average of 3 measurements) using NanoDrop One/OneC spectrophotometer: the exact molarity of each solution was obtained using software available online. RNA Split1 and Split2 sequences (8 nt, 4 nt or 0 nt) were mixed in a 1:1:1 ratio (0.12 μM) with complementary CP3 or with the negative controls PE or PR and incubated at 37 °C for 25 min in 20 mM Tris-HCl pH 7.5, 1 mM EDTA and 10 mM MgCl_2_ (hybridization buffer). To compare the hybridization of each Split1/Split2 system (8 nt, 4 nt or 0 nt) with target and non-target sequences, all samples (6.5 μL) were run on 10% Novex^TM^ TBE gel in 1× TBE buffer at 200 V for 40 min. After staining with SYBR^®^ Gold in 1× TBE for 5 min, the gel was visualized using a Typhoon laser scanner and Image Quant TL software (normal sensitivity and PMT 380 V, GE Healthcare Life Sciences). The low-range ssRNA ladder (NEB) was used as a marker of molecular weight.

For the in-gel imaging assay, Split1-4nt (0.12 μM) and Split2-4nt (0.12 μM) were mixed with target (0.12 μM, 0.10 μM, 0.08 μM and 0.05 μM of CP3) and non-target sequences (0.12 μM, PE or PR as negative controls). All samples (4.5 μL from 72 μL of reaction mix), including negative controls and the Broccoli aptamer (positive control, 25–30 ng), were run on 6% Novex^TM^ TBE gel in 1× TBE buffer at 180 V for 30 min. In-gel imaging was performed as described above (Section 3.3).

### 3.10. Assembly of RNA Split Sequences with Complementary CP3 and Non-Complementary Sequences PE and PR: PAGE and Fluorescence Measurements by qPCR Equipment

All the DNA and RNA sequences were prepared as described above. RNA Split1 and Split2 sequences (4 nt) were mixed in a 1:1:1 ratio (0.12 μM) with complementary CP3 or the negative controls PE or PR and incubated at 37 °C for 25 min in different buffers. The used buffers were as follows: (a) 40 mM HEPES, 100 mM KCl and 1 mM MgCl_2_ (DFHBI-1T buffer), (b) DFHBI-1T buffer supplemented with 5 MgCl_2_, (c) DFHBI buffer supplemented with 10 MgCl_2_, (d) 20 mM Tris-HCl pH 7.5, 1 mM EDTA and 10 mM MgCl_2_ (hybridization buffer), (e) hybridization buffer supplemented with 50 mM KCl and (f) hybridization buffer supplemented with 100 mM KCl. After the isothermal incubation, the resulting products were run on 10% Novex^TM^ TBE gel in 1× TBE buffer at 200 V for 40 min. After staining with SYBR^®^ Gold in 1× TBE for 5 min, the gel was visualized using a Typhoon laser scanner and Image Quant TL software (normal sensitivity and PMT 380 V, GE Healthcare Life Sciences). The low-range ssRNA ladder (NEB) was used as a marker of molecular weight.

The fluorescence measurements in liquid were conducted using the real-time PCR cycler Rotor-Gene Q (Qiagen) considering the following protocol: Split1-4nt (0.05 μM) and Split2-4nt (0.05 μM) were mixed in ice with target (0.05 μM, 0.04 μM, 0.03 μM, 0.02 μM and 0.01 μM of CP3) and non-target sequences (0.05 μM, PE or PR as negative controls) in 20 mM Tris, 1 mM EDTA, 100 mM KCl and 2.5 mM MgCl_2_.

After the addition of DFHBI-1T solution (final concentration 0.5 μM) in ice, all samples (35 μL), including negative controls and the Broccoli aptamer (positive control, 17 ng), were run in the real-time PCR cycler at 37 °C for 15 cycles (1 cycle/min) with excitation and emission at 470 nm and 515 nm, respectively (gain 10). The limit of detection (LOD) was calculated considering the standard deviation of the blank (10 x 3 independent measurements) and the calibration curve slope. All data are shown as mean ± standard deviation (SD), considering *n* = 3. For each measurement, 3 readings were acquired and averaged.

## 4. Conclusions

Here, we demonstrate the versatility of a Broccoli split aptamer system able to turn on the fluorescence signal of a non-cytotoxic and cell-permeable dye only in the presence of a specific RNA or DNA target sequence. We show that a designed three-way junction carrying the split system can be successfully expressed and assembled in a cell-free TX-TL system, paving the way to *in vivo* applications. Our results demonstrate the expression and assembly of a three-way junction restoring the functionality of the specific RNA ‘light-up’ aptamer in a reconstituted cellular environment. Accordingly, we underline the possibility of using a cell-free system as a technology to prototype functional RNA assemblies to be then expressed in synthetic cells or in bacterial cells. In this context, split fluorescent RNA aptamers can be used to detect specific hybridization or disassembly events as an alternative to Förster resonance energy transfer spectroscopy, unfeasible in a cellular environment when expressing specific sequences.

As the recognition arms can be accurately changed, the split system was redesigned, and its functionality was successfully demonstrated through the self-assembly of ‘bio-orthogonal’ hybrid RNA/DNA origami. We further confirm the versatility of our protein-free binary aptamer as a simpler reporter system compared with previously described approaches based on the catalytic hairpin assembly circuit [11] and *in situ* amplification method [13], in which the target is disconnected from the reporter system or may interfere with the cellular behaviour, respectively.

Finally, the RNA split aptamer sequences were elongated with a target recognition sequence specific to *Campylobacter* spp. After the selection of the Split1/Split2-4nt sequences and the optimization of the buffer composition, the split system was able to detect the femtomoles of a specific DNA target at 37 °C in less than 20 min. This emphasizes that our split system could be used in specific DNA sequence detection and considered a promising tool for future biosensor development as an alternative to molecular assays, such as real-time PCR with an LOD of 103 ng of DNA purified from meat samples [61]. Biosensors based on our split aptamer system do not require the detection of *Campylobacter* spp. pathogenic cells (specific live cells aptasensor) [62], overcome the DNA polymerase inhibition during PCR due to sample contaminants [61], and can allow a faster detection compared with the official ISO 10272:20 method used for *Campylobacter jejuni*, *C. coli, C. lari* and C. *upsaliensis* detection in food, which requires up to 7 days. To further optimize our system by reducing the background noise, future developments can focus on the following: (i) immobilizing a biotinylated split strand on streptavidin magnetic beads to introduce stringent washing steps normally used in dot blot assays, and (ii) adding destabilizing mismatches in the linker sequence of Split1-4nt and Split2-4nt, as previously mentioned by Kolpashchikov et al. [1].

Because the reaction occurs isothermally at a physiologically compatible temperature, our system can be genetically encoded, or the sensing reaction can be run where the traditional diagnostic polymerase chain reaction is unfeasible.

Overall, our results may find applications in the following areas: (i) labelling DNA data storage structures [19] to monitor multiple hybridization steps or disassemblies *in vitro* or *in vivo*; (ii) the *in vivo* tracking of tagged hybrid nanostructures used to deliver mRNA or siRNA as therapeutics; and (iii) new biosensing devices to detect specific DNA or RNA sequences.

## Figures and Tables

**Figure 1 ijms-24-08483-f001:**
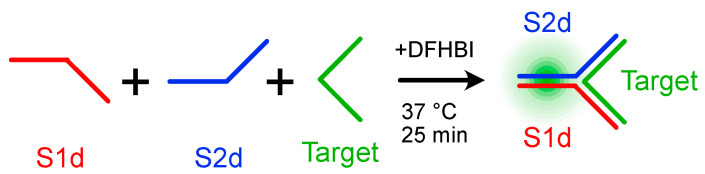
Schematic illustration of the self-assembly of the Broccoli three-way junction. Hybridization at 37 °C for 25 min of Split1d, Split2d and the complementary target sequence results in a 3-strand complex with the Broccoli aptamer. The complex assembly turns on the fluorescence in the presence of the specific dye DFHBI-1T.

**Figure 2 ijms-24-08483-f002:**
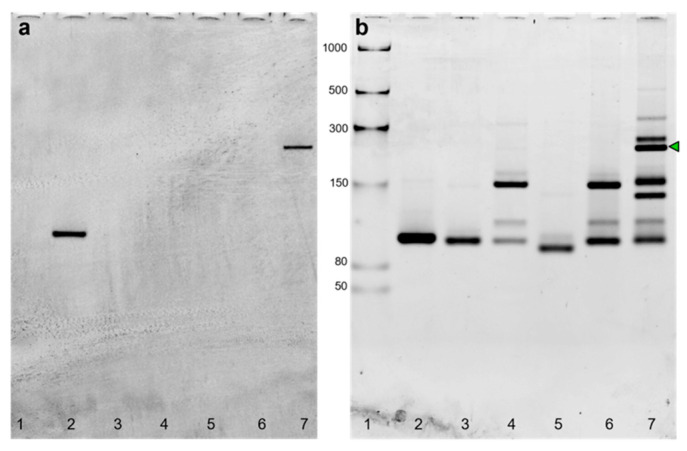
In-gel imaging of Broccoli aptamer and three-way junction hybridized *in vitro* from transcribed sequences. The 6% TBE gel electrophoresis after DFHBI-1T (**a**) and after SYBR^®^ Gold (**b**) staining. The gel was stained with DFHBI-1T for 3 min to visualize the Broccoli aptamer (positive control), and the hybridized Split1d, Split2d and target. After 3 washing steps, the gel was stained with SYBR^®^ Gold for 5 min. Lanes: 1: low-range ssRNA ladder; 2: Broccoli aptamer; 3: Split1d; 4: Split2d; 5: target; 6: Split1d and Split2d; 7: Split1d, 2d and target. The green arrow underlines the 3-way junction assembly. Molecular sizes in nucleotides are indicated.

**Figure 3 ijms-24-08483-f003:**
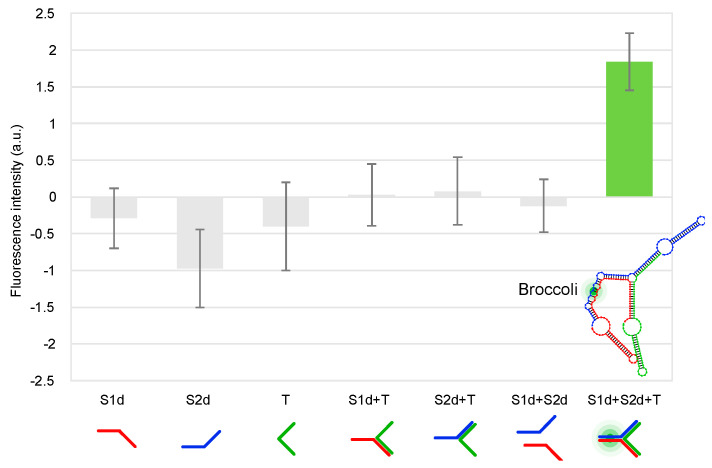
Fluorescence measurements in the PURExpress cell-free TX-TL system containing gBlock templates. Fluorescence emission intensity is expressed in a.u. (F-F0, where F = sample fluorescence intensity; F0 = cell-free system fluorescence without gBlock templates). From left to right: Split1d; Split2d; complementary sequence; Split1d and complementary sequence; Split2d and complementary sequence; Split1d and Split2d; Split1d, Split2d and complementary sequence (mean ± standard deviation, *n* = 3). All results were statistically analysed using ANOVA and two-tailed *t*-test; *p*-values of <0.01 are considered to be statistically significant.

**Figure 4 ijms-24-08483-f004:**
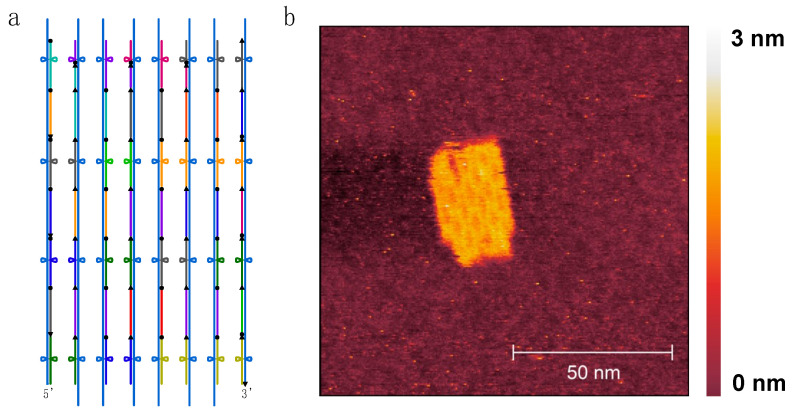
Scadnano schematic design (scaffold underlined in blue, staple strands shown in different colours) (**a**) and high-resolution AFM image (**b**) of hybrid rectangle RNA/DNA origami.

**Figure 5 ijms-24-08483-f005:**
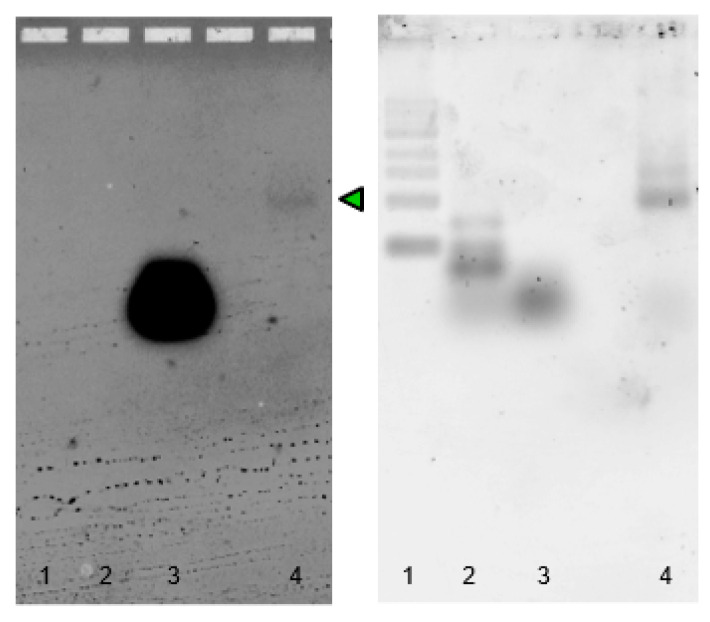
In-gel imaging of Broccoli aptamer and purified RNA/DNA hybrid origami. The 1.5% TAE agarose gel after DFHBI-1T (**left**) and after SYBR^®^ Gold (**right**) staining. The gel was stained with DFHBI-1T to visualize the Broccoli aptamer (positive control) and the purified hybrid RNA/DNA origami. After 3 washing steps, the gel was stained with SYBR^®^ Gold for 10 min. Lanes: 1: 1 Kb ladder; 2: low-range ssRNA ladder; 3: Broccoli aptamer; 4: purified hybrid RNA/DNA origami (underlined by a green arrow).

**Figure 6 ijms-24-08483-f006:**
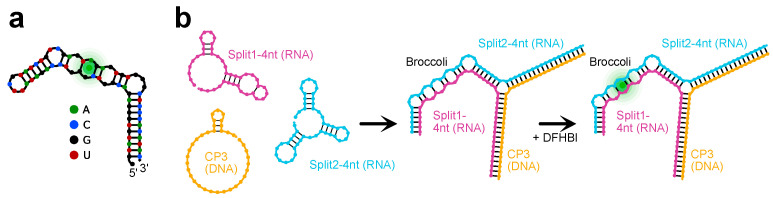
Split fluorescent Broccoli aptamer system to detect a selected DNA target from *Campylobacter* spp. (**a**) Secondary structure of the non-split Broccoli RNA aptamer sequence. (**b**) The hybridization of Split1-4nt, Split2-4nt and CP3 strands to yield a 3-strand complex with the Broccoli aptamer. Addition of DFHBI causes Broccoli fluorescence. Split1-4nt and Split2-4nt sequences were experimentally selected over other split sequences with shorter arms (as reported and discussed in Section 2.5).

**Figure 7 ijms-24-08483-f007:**
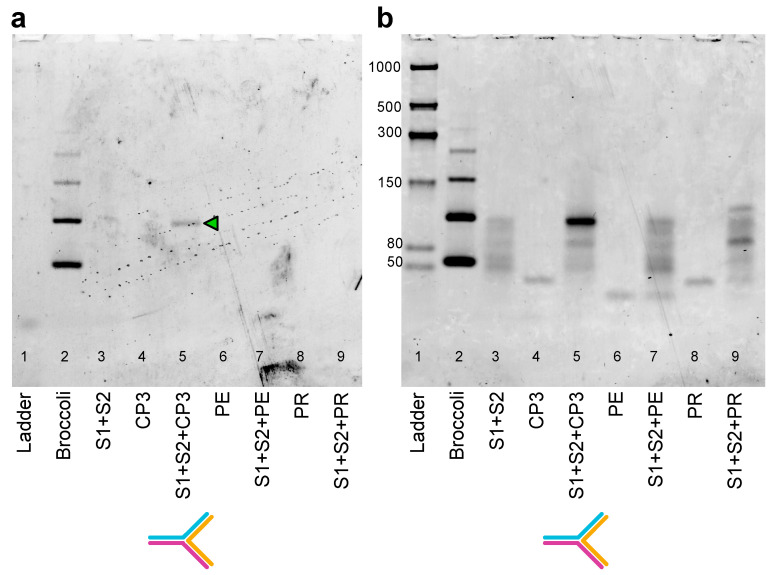
In-gel imaging of the split system in the presence of CP3, PE or PR sequences. The 6% TBE polyacrylamide gel after DFHBI-1T (**a**) and after SYBR^®^ Gold (**b**) staining. The gel was stained with DFHBI-1T for 4 min to visualize the Broccoli aptamer (positive control) and the hybridized Split1/Split2-4nt/CP3. After 3 washing steps, the gel was stained with SYBR^®^ Gold for 5 min. Lanes: 1: low-range ssRNA ladder; 2: Broccoli aptamer; 3: Split1/Split2-4nt (0.12 µM each); 4: target CP3 (0.12 µM); 5: Split1/Split2-4nt and target CP3 (0.12 µM each); 6: PE (negative control, 0.12 µM); 7: Split1/Split2-4nt and PE (0.12 µM each); 8: PR (negative control, 0.12 µM); 9: Split1/Split2-4 nt and PR (0.12 µM each). Molecular sizes in nucleotides are indicated; the green arrow indicates the reconstituted split Broccoli aptamer.

**Figure 8 ijms-24-08483-f008:**
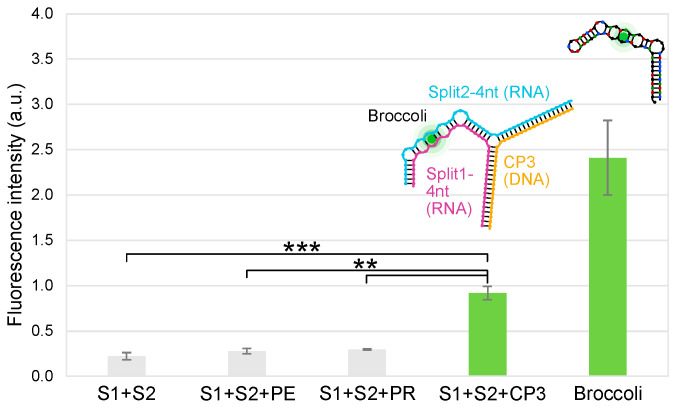
Fluorescence emission intensity in a.u. of the split system in the presence of an equimolar concentration of PE, PR or CP3 sequences. From left to right: Split1/Split2-4 nt, Split1/Split2-4nt/PE, Split1/Split2-4nt/PR, Split1/Split2-4nt/CP3 (0.05 µM equimolar concentration) and Broccoli (17 ng), (raw data are blank subtracted, mean ± standard deviation, *n* = 3). All results were statistically analysed using ANOVA and the two-tailed *t*-test: *** and ** indicate statistical significance with *p*-value < 0.001 and *p*-value < 0.01, respectively.

**Figure 9 ijms-24-08483-f009:**
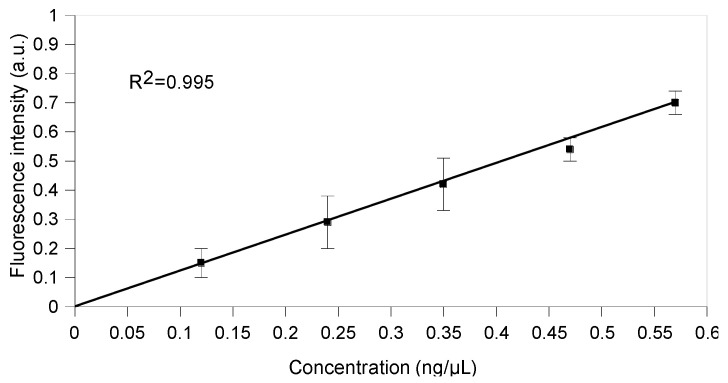
Sensitivity curve of the split system in the presence of different CP3 target concentrations. The fluorescence intensity in a.u. was obtained using Split1, Split2 (0.05 µM equimolar concentration) and different target CP3 concentrations from 0.12 ng/µL to 0.57 ng/µL (0.01 µM, 0.02 µM, 0.03 µM, 0.04 µM and 0.05 µM) (raw data values are background subtracted; mean ± standard deviation, *n* = 3). The sensitivity is the slope of the calibration curve, and the linear function is f(x) = 1.192x + 0.003.

## Data Availability

Not applicable.

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
