# Peer review of "Light-Up Split Broccoli Aptamer as a Versatile Tool for RNA Assembly Monitoring in Cell-Free TX-TL Systems, Hybrid RNA/DNA Origami Tagging and DNA Biosensing"

_ijms, 2023, doi:10.3390/ijms24108483_

Round 1

Reviewer 1 Report

I believe the paper presents interesting results and all-in-all the claims are well supported. Statistics are lacking, however.

Major

-Figure S4 would better demonstrate the correct formation of the broccoli aptamer if the fluorescence from the bound aptamer was shown in addition to the total staining.

Statistics are not shown for figure 1. Was the fluorescence higher in the positive system in a manner that is statistically significant?

Statistical significance also needs to be shown for figure 4.

Please consider citing and discussing: doi: 10.3390/molecules23123178; doi: 10.1021/acs.jchemed.7b00759.

Reviewer 2 Report

The authors presented a very interesting light-up nano-biosensor in vitro by integrating biotechnologies, split aptamer technology, and DNA/RNA nanotechnologies. The novelty is high; however, the quality of the presentation must be improved to increase the interest of the readers. Moreover, the manuscript is written very technically, and I recommend easing the technical words for easy reading. After the authors address the points below, I will be happy to recommend the manuscript for publication.

- Etc is not needed in the 2nd line of the introduction.

- I suggest to define "Broccoli aptamer" as simply "Broccoli" in the introduction, instead of using both names indistinctly. 

- 3rd paragraph of introduction: In "making the system unsuitable for stable nucleic acid self-assemblies (e.g., origami nanostructures).", the use of origami nanostructures is not clear for the reader. I suggest using "DNA origami [refs] and RNA origami [refs]" instead of "origami nanostructures", where you can reference relevant papers of aptamers integrated to DNA origami nanostructures. For instance, DNA origami: 10.1016/j.bios.2022.115053 and 10.1039/D0TB01291B, and RNA origami: 10.1093/nar/gkac470. You may also want to organize the references you provide in the manuscript and include them here.

- You may also want to reference other light-up systems based on aptamers and DNA origami. Some examples are in the above point, but the authors are welcome to include other references from their own search.

- Concatemeric instead of concatameric.

- If you say "On the other hand", you must first say "On one hand," somewhere else. They are like both half parts of a Broccoli aptamer: they go hand in hand. Please review the text.

- Is the sentence below a result of the authors in this manuscript? If so, why it is in the introduction?

When we considered the split system [11] as a component of a three-way junction with T7 terminators, the full Broccoli aptamer reconstitution was not successfully predicted.

- 4th paragraph of the introduction: Something does not sound good here: "and memory structures are considered as potentially generic interface to bacterial cell processes". I believe it should be: "as a potential generic"

- I believe using coll. to say collaborators is not standard (but I may be wrong here). It was quite confusing the first time I saw coll. in this manuscript. I strongly recommend saying the full word; for instance,  "Filonov and collaborators" or "Filonov and coworkers" or the shorter "Filonov et al." Regardless of the choice of the authors, please keep the choice consistent throughout the text.

- In general, for the introduction/presentation to your work, a single paragraph clearly and simply communicating the main hypothesis and/or objectives is strongly required.

- The authors do not need to put this sentence there: "In detail, the split sequences, ending with 8 nt of a stabilizing bioorthogonal scaffold,[8] were elongated in 5’ and 3’, while the 4 nt loop UUCG was removed from the stem.[13,14]" You can put that part in either the methodology, the presentation to the results, or in the introduction to the manuscript.

- In "self-assembly of a well-defined rectangle shape (40 nm x 25 nm)," you have the opportunity to define the "hybrid RNA/DNA origami". For instance, "self-assembly of a well-defined rectangle shape (40 nm x 25 nm) hybrid RNA/DNA origami," 

- The authors do not need "Finally" in "Finally, we demonstrate the application...", if you have a subsequent paragraph. It looks strange to have finally. 

- Fig. 2: The units in the AFM colorbar are not visible, i.e., the m in nm is not visible.

- Can the authors elaborate more on this “probably due to the extension orientation with respect to the scanning direction.[33]”? because it is not clear

- It is surprising that the aptamer is not visible in AFM, and not expected to see an effect of the AFM scan direction on the height of the sample. Normally, it would be expected to see a "bump" unless the tethered particle has rotational freedom on the surface of the origami. The original DNA origami paper by Rothemund shows origamis with hairpins that are visible in AFM. It might be the case that the conformation of the aptamer in the current manuscript (flexible because it is composed of flexible segments), is much more flexible than the hairpins. The authors should comment on this idea in their paper.

- In methodology: It is recommended the authors write in a line(s) what the procedure of Kozyra et al. [27] for sequence design is about.

- In methodology: What is the implication of simulating the RNA structures with high NaCl concentration on the experimental design with low ionic concentrations? Can the authors elaborate on this? In other words, how much does the simulation tell about the structures at differing concentrations of MgCl2 and KCl? I believe this discussion is important to make a stronger argument for the potential applications presented in the las paragraph of the Conclusions section.

- In Fig. 3, the authors provide the RNA simulation showing the bases but the thermodynamic result from NUPACK is not shown. Can the authors provide in SI the simulated structure of the full RNA complex?

- Can the authors describe in the Main Text or Supplementary information the multiple bands (at least three bands) in lane 5 of Fig. S6. Is the fast band indicating the scaffold? Or a misfolded structure? What about the slow band?

- Can the authors define in a few lines de Bruijn sequence? Also, how the authors exactly use the knowledge of the reference publications in the current manuscript to determine the sequences.

- Can the authors comment about the purity of the “RNA scaffold”? This is if what they got as RNA sequence is what they expect. List S2 gives the DNA sequence that was inserted, but there is no information about the sequence that was purified.

- Can the authors provide the linear function for the fitting in Fig. 5.

- The authors also mentioned that their system was demonstrated “suggesting a biocompatible alternative to organic dyes for in vitro and in vivo applications.” Can the authors elaborate why is this the case? Is there any particular organic dye and system which they can refer to. Maybe talk about the quantum yield of their system. Also, for in vivo applications, what would be a system to compare their system to?

- Fig. S22: it looks interesting the bended band artifact in the fast band of lane 2. lane 8, and others. Can the authors explain in the figure why it happened among the three gels?

Reviewer 3 Report

In this manuscript, the authors reported biosensors based on Broccoli aptamer for RNA assembly monitoring. This work is interesting. Specific comments are as follows:

1. As the conventional performances of analytical method, the selectivity, the stability and real sample assay of this assay method should be added.

2. The schematic diagram of this assay method should be added.

3. Comparison of this assay method with the other detection methods should be discussed to highlight the advantages of this approach. It would be nice if the author could make a list.

4. In introduction, the most reference is published before 2021. Several recent papers about RNA or DNA biosensors (such as Biosensors and Bioelectronics, 2022, 206: 114120) should be cited.

5. The statement about the universal background is too long and should be refined in introduction. Authors should highlight the innovation point.

6. Considering authors highlight that the assay method can be used for specific nucleic acid sequence detection, authors should explain why the method can avoid mismatching.

Reviewer 4 Report

Light-up aptamers are an emerging class of aptamers that can enhance the fluorescence of different dyes upon their specific binding. These aptamers  could be potentially used for monitoring of different RNA- and DNA-related processes in living cells or development of aptasensors for DNA detection. In the present paper authors proposed split Broccoli aptamer consisting of two independent RNA molecules that could form full-active light-up aptamer during binding of specific organic dye (DFHBI). The proposed binary Broccoli aptamer was used for monitoring of its self-assembly in an E. coli transcription-translation system. Authors also showed the application of the split Broccoli aptamer system for the sensitive in vitro detection of a DNA sequence from the Campylobacter spp. So, this study is very important both in fundamental and practical molecular biology.

The presented paper is very well written, all the results are thoroughly illustrated and clear. However, I have some comments and questions I would like to be answered in the paper. I believe it could further improve this manuscript.

1.      I think that it would be much better for adequate understanding of the results if some figures from Supplementary materials were transferred to the main manuscript. For example, Figure S2 would be very useful in the section 2.1. It would be nice to add a scheme illustrating the whole split aptamer formation strategy in this section. The same is true for figure S11.

2.      Why lane 7 at the Figure S4 looks like it is a part of another gel? Was this line a part of original gel or was it added from another gel?

3.      In the section 2.3 an additional scheme of triple complex between different variant of split Broccoli aptamers and DNA target will make it easier to understand the whole strategy.

4.      At the figure 3с and d (lane 2) control Broccoli aptamer presented by 4 different bands. At the same time control Broccoli aptamer at the Figure S2 (lane 2) presented by single band. What could be the possible explanation of this difference?

5.      At the capture of the Figure S19 concentrations of CP3 target presented using μM, but at the page 7 of the manuscript the lower detected concentration of CP3 referred as ng/μL. It would be clearer to use the same concentration units throughout the whole text.

Round 2

Reviewer 2 Report

My points have been addressed.

Reviewer 3 Report

Authors appropriately responded to the questions and comments raised by reviewers. I believe that authors have carefully revised their paper. So, I think it acceptable for publication.